

# THE PROPAGATION OF INVENTORY-BASED POSITIONAL ERRORS INTO STATISTICAL LANDSLIDE SUSCEPTIBILITY MODELS

Stefan Steger[1], Alexander Brenning[2], Rainer Bell[1], and Thomas Glade[1]

1 Department of Geography and Regional Research, University of Vienna, Universitätsstraße 7, A-1010 Vienna, Austria
Department of Geography, Friedrich Schiller University Jena, Loebdergraben 32, D-07743 Jena, Germany

*Correspondence to: Stefan Steger (stefan.steger@univie.ac.at)*

**Abstract.** There is unanimous agreement that a precise spatial representation of past landslide occurrences is a prerequisite to produce high quality statistical landslide susceptibility models. Even though perfectly accurate landslide inventories rarely exist, investigations of how landslide inventory-based errors propagate into subsequent statistical landslide susceptibility models are scarce.

The main objective of this research was to systematically examine whether and how inventory-based positional inaccuracies of different magnitudes influence modelled relationships, validation results, variable importance and the visual appearance of landslide susceptibility maps. The study was conducted for a landslide-prone site located in the districts of Amstetten and Waidhofen/Ybbs, Eastern Austria, where an earth-slide point inventory (n = 591) was available.

The methodological approach comprised an artificial introduction of inventory-based positional errors into the present landslide data set and an in-depth evaluation of subsequent modelling results. Positional errors were introduced by artificially changing the original landslide position by a mean distance of 5, 10, 20, 50 and 120 m. The resulting differently precise response variables were separately used to train logistic regression models. Odds ratios of predictor variables provided insights into modelled relationships. Cross-validation and spatial cross-validation enabled an assessment of predictive performances and permutation-based variable importance. All analyses were additionally carried out with synthetically generated data sets to further verify the findings under rather controlled conditions.

The results revealed that an increasing positional inventory-based error was generally related to increasing distortions of modelling and validation results. However, the findings also highlighted that interdependencies between inventory-based spatial inaccuracies and statistical landslide susceptibility models are complex. The systematic comparisons of 12 models provided valuable evidence that the respective error-propagation was not only determined by the degree of positional inaccuracy inherent in the landslide data, but also by the spatial representation of landslides and the environment, landslide magnitude, the characteristics of the study area, the selected classification method and an interplay of predictors within multiple variable models. Based on the results, we deduced that a direct propagation of minor to moderate inventory-based positional errors into modelling results can be partly counteracted by adapting the modelling design (e.g. generalization of input data, opting for strongly generalizing classifiers). Since positional errors within landslide inventories are common and subsequent



modelling and validation results are likely to be distorted, the potential existence of inventory-based positional inaccuracies should always be considered when assessing landslide susceptibility by means of empirical models.

**Keywords**: Landslide susceptibility, Logistic regression, Landslide inventory, Validation, Variable importance, Error propagation;

## 5  1. Introduction

The analysis of landslide susceptibility has been a highly active research topic during the past decades (Wu et al., 2015). The term landslide susceptibility refers to the likelihood of a certain location to be affected by upcoming landslide events (Brabb, 1984; Corominas et al., 2013; Fell et al., 2008; Guzzetti et al., 1999, 2005).

For large regions or areas where detailed geotechnical information is missing, statistically-based approaches are most
commonly applied to generate landslide susceptibility maps (Cascini, 2008; Corominas et al., 2013; van Westen et al., 2008). The underlying concept is based on a modified uniformitarian principle (Hutton, 1788) by assuming that future landslides evolve more likely under those environmental conditions that led to past slope movements (Brabb, 1984; Ermini et al., 2005; Fabbri et al., 2003; Fell et al., 2008; Glade et al., 2005).

In practice, this idea is operationalized by fitting a binary classification model to a data set containing spatial information on
the presence and absence of past landslides (response) and a number of associated static preparatory environmental factors (predictors; e.g. slope, lithology). The resulting classification rule allows to identify conditions that were likely to promote past slope movements, and conditions that favoured slope stability. When the constructed model is considered to appropriately describe the underlying relationship, a likelihood of landslide occurrence can be assigned to all spatial units containing information on these environmental conditions. Thus, a specific location shows a high landslide susceptibility whenever the
respective environmental setting is similar to the conditions observed for past landslide occurrences (Ermini et al., 2005; Goetz et al., 2015b; Petschko et al., 2014a; Van Den Eeckhaut et al., 2006).

The quality and applicability of a landslide susceptibility map is regularly inferred from predictive performance estimates which compare the predicted susceptibility pattern with an independent data set that is supposed to mimic future landslide locations (Beguería, 2006; Brenning, 2005; Chung and Fabbri, 2003; Frattini et al., 2010; Remondo et al., 2003). An issue
found by many authors is that the quality of the presented landslide susceptibility map is directly related to the quality of the input data used to construct the underlying model (Cascini, 2008; Corominas et al., 2013; Fressard et al., 2014; Guzzetti et al., 2006; van Westen et al., 2008). In this context, an accurate representation of past landslide activities (landslide inventory) is regularly considered as a crucial component to produce reliable results (Ardizzone et al., 2002; Fressard et al., 2014; Galli et al., 2008; Guzzetti et al., 2012; Harp et al., 2011; Hussin et al., 2016; Steger et al., 2015, 2016). However, perfect landslide
inventories are known to be rarely available (Malamud et al., 2004; Petschko et al., 2015). Several studies emphasize that the positional accuracy of a landslide inventory can vary greatly and is influenced by the type (e.g. aerial photograph, shaded-relief image) and quality (e.g. spatial resolution) of available base maps, the type and sizes of mapped landslides, human factors



(e.g. accuracy of public reports, covered time period) and the processing of analogue information sources (e.g. digitization) (Ardizzone et al., 2002; Harp et al., 2011; Petschko et al., 2015; Santangelo et al., 2015).

A number of authors compared landslide susceptibility models generated from different inventories (e.g. Ardizzone et al., 2002; Fressard et al., 2014; Galli et al., 2008; Steger et al., 2015, 2016; Zêzere et al., 2009). However, differences between the respective landslide inventories could only be approximated in relative terms due to a variety of underlying data sources and an unavailability of quantitative information on the specific positional errors. Thus, specific statements on how a particular positional error may propagate into the final models and maps could not be made.

This study aims to shed more light on the impact of positionally erroneous landslide inventories on the results of statistical landslide susceptibility models by artificially controlling for the extent of positional inaccuracy inherent in the respective data set. The main objective was to systematically examine the influence of positionally inaccurate landslide inventories on modelled relationships, validation results and the appearance of landslide susceptibility maps. All analyses were also conducted with synthetically generated data to further validate the results under controlled conditions.

## 2. Study area

The study site extends over 100 km² (20 km x 5 km) and is located in the eastern part of Austria (Fig. 1A). The area belongs to the Lower Austrian districts Amstetten and Waidhofen/Ybbs. Altitudes range from less than 350 m a.s.l. in the northern valley bottoms to 790 m a.s.l. at the southern hilltops. The landslide-prone area (5.9 landslides / km²) can be characterized as a hilly undulating landscape with an average slope gradient of 11° (Fig. 1).

The morphological and lithological condition of the study site plays a major role for the propensity of the respective slopes to fail. The major part of the area (81 km²) relates to alternating sediment sequences of the Rheno-Danubian Flysch Zone (Fig. 1B). Several publications emphasize that the high density of shallow landsliding within the Flysch is related to the widespread presence of deeply weathered soils consisting of silts and clayey material (Petschko et al., 2015; Schwenk, 1992; Wessely et al., 2006). A minor portion of the terrain (3 km²) can be assigned to clastic sediments of the Molasse Zone. The Molasse lies along the northern side of the Eastern Alps and is regularly associated with several meter thick clayey and marly soils which tend to be susceptible to landsliding, even at lower slope gradients (Wessely et al., 2006; Petschko et al., 2015). Quaternary sediments (16 km²) that relate to alluvial deposits and fluvial terraces dominantly cover the valley floors of the area. Landslide density is low within this area, mainly due to its prevalent flat topography (compare Fig. 1B with Fig. 1C).

In the light of the general litho-morphological predisposition of the area, landslide triggering is primarily related to a direct water-influx into the subsurface. Most landslides of the area occurred after intensive precipitation and snow-melting events (Petschko et al., 2015; Schweigl and Hervás, 2009; Schwenk, 1992). The resulting reduction of internal friction and increase in water-related loading are considered to be the main driving forces along a regularly distinct sliding plane. However, also human interventions were observed to be an increasingly important landslide influencing factor (Schwenk, 1992; Wessely et al., 2006). The southern lying hilly areas are for the most parts used for intensive cattle farming (e.g. pastures) or covered by





forests while arable land is prevalent in the northern lying flat lowlands (Eder et al., 2011). The area lies in a transitional climate zone (oceanic influences from the West and continental influences from the East) with mean annual precipitation rates around 1,000 mm (Skoda and Lorenz, 2007).

The majority of landslides in the area are relatively small and shallow. Petschko et al. (2015) investigated landslide sizes for the districts Amstetten, Waidhofen/Ybbs and Baden and found a median size of landslides of the slide-type movement of 787 m² for the Rheno-Danubian Flysch Zone, 1,189 m² for the Molasse Zone and 1,260 m² for Quaternary sediments.

## 3. Data

### 3.1 Landslide inventory

The historical landslide inventory used for this study represents a subsample of landslides mapped by Petschko et al. (2015). The original data set consists of 13,166 point features which represent the initiation zone of landslides of the slide-type movement (one point per landslide) and was compiled by analyzing mainly topographic derivatives of an airborne laser scanning (ALS) digital terrain model (DTM; 1 m x 1m), supported by interpreting orthophotos. The point-inventory was generated for susceptibility assessments while also several other studies endorse to use one point per landslide observation (Atkinson et al., 1998; Goetz et al., 2015a; Petschko et al., 2014a; Van Den Eeckhaut et al., 2006). The sample used for this study consists of 591 landslides. Further information on the mapping of this inventory and its related advantages and disadvantages are described by Petschko et al. (2015).

### 3.2 Predictor variables

The number and type of predictors used within statistical landslide susceptibility analyses varies greatly depending on the scope of the study, the type of the investigated landslides, the characteristics of the area and data availability (Guzzetti et al., 1999; van Westen et al., 2008). To enhance traceability of modelling results, we opted to include only few but widely used predictors within the analyses. All models were generated with a predictor set consisting of a lithology layer (Fig. 1B) and three topographic variables, namely slope (Fig. 1C) and two variables representing slope orientation (Fig. 1D,E).

The inclination of a slope is directly related to shear stresses and therefore almost always used as a predictor within statistical landslide susceptibility modelling. Slope aspect is regularly introduced as a proxy for a varying degree of insolation and evapotranspiration influencing moisture and weathering conditions or, in some cases, as a representative for the dipping of geologic structures (Gorsevski et al., 2006; van Westen et al., 2008). The two slope azimuth predictors were directly derived from a resampled (10 m x 10 m) ALS-based DTM. Classification of the continuously scaled aspect layer (originally ranging from 0° to 360°) was avoided by calculating its cosine (representing the degree of north-exposedness) and sine (east-exposedness) (Brenning, 2009, 2012a). This transformation ensured that the applied linear models treated similarly oriented slopes (e.g. 1° vs. 360°) similarly. At regional scales, the parent material of the soils is usually represented by lithological maps



(Gorsevski et al., 2006; van Westen et al., 2008). This information was obtained from a digital geological map of Lower Austria (GK200) and resampled to the modelling resolution of 10 m x 10 m.

We decided to exclude land cover from the analysis for two reasons. Firstly, the available recent land cover data does not necessarily correspond to the land cover present at the time of landslide occurrence, due to constantly ongoing land use changes

(Petschko et al., 2014b; van Westen et al., 2008). Secondly, an omission of land cover as a predictor was expected to reduce the chances that the suspected land cover related-bias (e.g. overrepresentation of landslides in forested areas, cf. Bell et al., 2012; Petschko et al., 2015) is directly propagated into the final results (cf. Steger et al., 2016).

## 4. Methods

In this study we aimed to analyze all data in a tangible and traceable way. Therefore, we avoided using less interpretable

classifiers or a high number of predictors. For classification we opted for the well-established logistic generalized linear model, also known as logistic regression (cf. Sect. 4.2), while we included only a small number of frequently applied predictors (cf. Sect. 3.2). The general methodological framework consisted of an artificial introduction of positional inaccuracies into the present landslide inventory (cf. Sect. 4.1) and a subsequent in-depth evaluation of modelling results. The final response variable included within each model was based on an equal number of landslide presence-observations (landslide inventory) and

randomly sampled absence-observations (1:1 sampling) (Goetz et al., 2015a; Heckmann et al., 2014). All analyses were additionally performed with synthetic data in order to further verify the observations under rather controlled conditions (cf. Sect. 4.6).

### 4.1 Artificial introduction of positional errors into the landslide inventory

Inventory-based positional errors were introduced by artificially changing the original landslide scarp position according to a

normal distribution in a random direction (Fig. 2). The mean distance to the original position was specified as 5, 10, 20, 50 and 120 m (Fig. 2C). Smaller positional inaccuracies (e.g. 5 or 10 m) may be present when digitizing landslide inventories mapped through interpretation of aerial photographs (Santangelo et al., 2015) while the highest value (120 m) was defined according to an analysis made for the Flysch Zone. In this case, 65 of the 681 analyzed landslide database entries of the Building Ground Registry (more information refer to Steger et al., 2016) contained a quantitative estimate on the positional

accuracy of the respective reported landslide (mean positional error: 120 m, standard deviation: 84 m).

### 4.2 Logistic regression and odds ratios

Logistic regression is probably the most widely used classifier to generate landslide susceptibility maps (Wu et al. 2015). Many studies confirmed its high suitability for the analysis of landslide susceptibility due to its ability to predict a binary response variable (absence and presence of landslides) using both, continuously and categorically scaled predictors while ensuring a

high generalizability, interpretability and smooth prediction surfaces (Atkinson and Massari, 1998; Felicísimo et al., 2013;





Goetz et al., 2015a; Steger et al., 2016; Van Den Eeckhaut et al., 2006). Logistic regression models of this study were fitted by using the R package "stats" (R Core Team, 2014). The probability of a landslide occurring $P(Y = 1)$ was predicted by:

$$logit(P(Y = 1)) = \beta_0 + \beta_1 X_1 + \cdots + \beta_p X_p \qquad (1)$$

in which, $\beta_0$ represents the intercept and $\beta_1 \ldots \beta_p$ the regression coefficients of predictors $X_1 \ldots X_p$. The associations between predictors and the response within multiple variable regression models can be expressed as odds ratios (OR). In contrast to an interpretation of probabilities, OR allow to express this relation with a single number while accounting for the influence of other predictors (Brenning et al., 2015; Hosmer and Lemeshow, 2000). For instance, OR estimated for the predictor lithology

10   display the estimated chances that a certain lithological unit is affected by landsliding while accounting for the possible confounding effect of slope angle (Steger et al., 2016).

The odds of an event occurring (in this case a landslide) are defined as the probability of this event occurring $P(Y = 1)$, divided by the probability that this event is not occurring $P(Y = 0)$. The ratio between the odds after a one-unit increase in the predictor $X_p$ and the original odds is referred to as odds ratio:

$$OR\left(X_p\right) = \frac{Odds(X_p + 1)}{Odds(X_p)} = \frac{Odds\ after\ a\ one\ unit\ increase\ in\ the\ predictor}{Original\ odds} \qquad (2)$$

For logistic regression based models, OR can be directly derived from exponentiated regression coefficients ($\beta_1 \ldots \beta_p$). OR of > 1 point to a positive relation between a continuously scaled predictor and landslide occurrence while ORs of < 1 express a

20   negative association. Within this study, OR obtained for all continuously scaled predictors were estimated for "meaningful" increments (Brenning, 2012a; Brenning et al., 2015; Hosmer and Lemeshow, 2000). For example, ORs presented for the predictor slope angle refer to the changes in the odds for a slope angle increase of $10°$ ($X_p + 10$). OR estimated for categorically scaled predictors have to be interpreted in relation to a specified reference level.

### 4.3 Comparison with reference models

25   An expert-based evaluation of the final results was conducted by comparing all modelled relationships and maps with the results of those models that were assumed to be less affected (i.e. reference model for the real data) or unaffected (i.e. reference for the synthetic data) by inventory-based positional errors. Consequently, we determined that a model and the consequent map may be strongly affected by an inventory-based inaccuracy, whenever the underlying modelled associations (i.e. OR of predictors) respectively the spatial pattern of the maps differed substantially from the respective reference models.





### 4.4 Assessment of the predictive performance

The prediction skills of the present models were estimated by calculating the AUROC by applying two partitioning techniques, implemented in the R package "sperrorest" (Brenning, 2012b), namely k-fold cross validation (CV) and k-fold spatial cross validation (SCV). CV consists of a repeated non-spatial random partitioning of the original sample into k-subsamples. Within

each iteration, a performance measure (e.g. AUROC) is estimated for one of the k-subsamples while the remaining sample (k-1) is used to fit the model. In contrast to CV, the partitioning within SCV is performed spatially and consists of a repeated spatial partitioning of the training sample and test sample (Ruß and Brenning, 2010). In the context of landslide susceptibility modelling, SCV can be applied to estimate the spatial transferability of modelling results (Goetz et al., 2015a; Petschko et al., 2014a). In this study, CV and SCV were repeated 50 times with 10 folds per repetition.

Furthermore, the AUROC was also used to quantitatively compare the ability of all models to differentiate landslide presences and landslide absences of the unmodified response variables (= goodness of model fit for models generated with the unmodified data set).

### 4.5 Estimation of predictor importance

Permutation-based variable importance assessments assume that the importance of a specific predictor is directly related to the

decrease in classification accuracy or predictive performance after randomly reordering (i.e. permuting) the values of this predictor (Strobl et al., 2007). In landslide susceptibility modelling, permutation-based variable importance was previously applied by Goetz et al. (2015a).

In this study, a predictor importance evaluation was conducted to assess whether and how the apparent importance of a specific predictor changes when the positional accuracy of the inventory changes. Thus, the AUROC decrease of each predictor was

assessed on non-spatial (CV) and spatial cross-validation partitions (SCV) using the R package "sperrorest" (Brenning, 2012b; Ruß and Brenning, 2010). Herewith, each predictor was permuted 50 times within each fold leading to a total number of 25,000 permutations per predictor (50 repetitions times 10 folds times 50 permutations) within each partitioning technique (CV and SCV) for each of the 12 models.

### 4.6 Generation of synthetic data

None of the susceptibility models generated for the present study area can be considered to reflect a true and unbiased relation between landslides and environmental conditions, especially due to the unavailability of perfect and accurate spatial information (i.e. landslide data and predictors). Thus, the conducted comparison of a reference model with models generated with erroneous inventories may only provide an indication of the effect of inventory-based inaccuracies on modelling results. Thus, all analyses were additionally performed with synthetically generated data. This procedure allowed us to define a "true"

association between landslides and their predisposing factors while further controlling for environmental conditions (e.g. spatial distribution and interrelations between predictors) and model specific parameters (e.g. sample size).



The preparation of synthetic data consisted of three major steps: (i) designing the environmental conditions of the study area (= predictors), (ii) specifying a "true" relation between landslides and predictors and (iii) simulating a landslide distribution within the study area according to the predefined "true" relation.

The areal extent of the synthetic study site (20 km x 5 km) was adopted from the Lower Austrian study area while a
topographically diverse terrain was generated by generalizing an already available DTM (cf. 3D image in Fig. 3A). The topographic predictors slope (Fig. 3A) and both aspect layers (Fig. 3B,C) were directly derived from this smoothed DTM, which was expected to represents undisturbed pre-failure conditions (cf. Van Den Eeckhaut et al., 2006). Lithology and land cover layers were generated by systematically intersecting and reclassifying randomly generated raster files produced at different spatial scales. The resulting pixilated grid files were smoothed and resampled (Fig. 3D,E).
The three lithological classes (33 km² each) were positioned in the Western (unit A) and Eastern half (unit B) of the study site, while unit C was similarly spread across the area (Fig. 3D). The spatial distribution of land cover classes (33 km² each) was conditioned on the spatial distribution of slope inclinations as regularly observed within European regions (Rickli et al., 2002; Steger et al., 2016; Van Den Eeckhaut et al., 2006). More specifically, we defined that the flattest areas (1st slope tercile) consist of arable lands (66%) and pastures (33%) while the steepest parts (3rd slope tercile) are dominated by forests (66%)
and pastures (33%). Only medium inclined slopes (2nd slope tercile) were equally covered by all three land cover classes. Landslide susceptibility was defined to be solely dependent on five factors, namely slope, northness, eastness, lithology and land cover (Tab. 1). For the definition of the effect of topographic variations, we oriented ourselves on observed associations for the Lower Austrian study area. Pastures and Arable lands were defined to be equally prone to landsliding (OR: 1) while the stabilizing effects of forests on shallow landslide activity was expected to decrease the chances of landsliding (OR: 0.5).
For the lithological units we specified that the units A and B should be equally prone to landsliding (OR: 1) while unit C was defined to be less susceptible (OR: 0.5).

Relationships depicted in Table 1 (first row) were back-transformed to regression coefficients while a rare events correction of the intercept (King and Zeng, 2001) was conducted to consider the low portion of landslide-presences in the area (~ 0.002 % of the cells were envisaged to represent landslide initiation) in relation to the 50 % of presence-data in the final response
variable (1:1 sampling strategy). The resulting logistic regression equation was then transferred to each raster cell of the area to obtain a probability raster expressing the predefined associations (Fig. 3F). A spatially balanced landslide data set was generated according to Theobald et al. (2007). This procedure allowed us to favor particular sampling locations according to an inclusion probability raster. The resulting sample (n = 2000; Fig. 3F) was considered to represent an unbiased and accurate landslide inventory and used as presence data within all previously mentioned analyses (cf. Sect. 4.1 to 4.5). The similarity of
prescribed relationships (OR in the first row of Tab. 1) and model estimates (OR in the second row of Tab. 1) indicated that the previously established associations were successfully describable by the logistic regression model fitted with artificially generated data sets (i.e. response variable and predictors). Thus, we further considered this model as a useful reference that reflects the "true" relation between the response and the respective environmental conditions.



## 5. Results

A first inspection of real and synthetic data sets revealed that landslides were more frequent on flatter slopes and less frequent on steeper slopes with a decreasing positional accuracy of the respective inventory (Fig. 2B, 3G). On average, slope angles of the original landslide locations were 4.1° (real data) and 4.6° (synthetic data) higher compared to the most inaccurate inventory

(mean error: 120 m). Landslide densities within lithological units changed comparably little when introducing the highest positional error (mean 120 m) into the inventories. Landslide densities within the Flysch decreased from 7.2 to 7 landslides per km² while landslide densities observed for the Molasse remained unchanged (1 landslide per km²). An increasing landslide density was detected for the Quaternary unit by introducing a mean positional error of 120 m (from 0.4 to 0.7 landslides per km²). The following changes in landslide densities were observed for the lithological units of the synthetically generated data

set by introducing a high positional error (mean: 120 m): The number of landslides per km² for unit A (respectively B) decreased from 33.3 (B: 14.1) to 31.4 (B: 12.7), while landslide density within the unit C increased from 13.3 to 14.9.

### 5.1 Modelled relationships

Modelled relationships obtained for the predictors slope angle, northness and eastness (for both data sets) and land cover (only synthetic data) indicated that the strength of association with landslide occurrence generally decreased with an increasing

positional error of the inventory since the respective OR tended to approach the neutral value of 1 (Fig. 4A,C,D,F,G). Only the predictor lithology (Fig. 4B,E) did not show this trend.

Modelled associations remained nearly unchanged when simulating the smallest positional error (mean: 5 m) (Fig. 4). OR changes became noticeable for a simulated mean positional error of 10 m (real data) respectively 50 m (synthetic data; compare OR with the dashed line in Fig. 4). Especially, OR of the predictors slope (Fig. 4A,D) and lithology (Fig. 4B,E) provided

quantitative evidence that the models generated with real data exhibit a higher sensitivity to medium positional errors of the inventory (e.g. mean: 10 m, 20 m) compared to the apparently more robust synthetic models.

OR further exposed that a decreasing accuracy of landslide inventories was related to decreasing sensitivity of modelling results on slope angle (Fig. 4A,D). For instance, the susceptibility model generated with real data and with the unmodified inventory showed that the chances of a 10° steeper slope to be affected by future landsliding are 7.3 times higher compared to

its 10° flatter counterparts. The respective model generated with the most inaccurate inventory (mean error: 120 m) predicted that the chances of landslide occurrence solely rises by a factor of 1.9 for such a slope angle increase (Fig. 4A). Nevertheless, each produced model still predicted a positive relation (OR > 1) between landslide occurrence and the inclination of a slope. OR obtained for the lithology layers revealed major changes for the Molasse (real data) respectively the lithological unit B (synthetic data) (Fig. 4B,E). The chances that those units will be affected by future landslides were modelled to be considerably

lower (i.e. most distant from the threshold of 1) whenever the respective model was generated with the most inaccurate data sets.





## 5.2 Visual appearance of maps

A visual inspection of landslide susceptibility maps generally confirmed that the respective modelled relationships were visually recognizable in the predicted susceptibility patterns. Thus, an increasing positional error of the inventory was observed to result in susceptibility maps that were less influenced by local topographic variations. Ultimately, this led to more uniformly

appearing susceptibility patterns at slope scale (e.g. Fig. 5F). Flatter slopes were generally predicted as more susceptible and steepest slopes as less susceptible with an increasing positional inventory-based error (Fig. 5). The declining effect of local slope variations was accompanied by an increasing visibility of lithological transitions. Especially the model generated with real data and the most inaccurate inventory (Fig. 5F) depicted this tendency by strongly accentuating the silhouettes of the Flysch Zone (cf. Fig. 1B).

## 5.3 Predictive performance

Predictive performances decreased with a decreasing positional accuracy of the inventories (CV and SCV in Fig. 5). However, models generated with a mean inventory-based error of 50 m still produced AUROCs of 0.76 (CV; real data) respectively 0.83 (synthetic data). Non-spatially assessed median AUROCs were observed to decrease from 0.85 (real data) and 0.86 (synthetic data) to 0.69 (real data) and 0.75 (synthetic data) (CV in Fig. 5). The general trend of decreasing prediction skills with an

increasing positional error was exposed by both partitioning techniques, namely CV and SCV. In analogy to observations of the modelled relationships, we observed a first distinct performance drop (AUROC decrease of > 0.1) for the models generated with an inventory-based positional mean error of 10 m (real data) respectively 50 m (synthetic data).

A comparison of predicted susceptibility patterns with the respective unmodified response variable revealed that all models performed well to forecast the original landslide position ("0" in Fig. 5). It was remarkable that models generated with highly

inaccurate inventories (mean error: 120 m) achieved AUROCs of 0.84 (real data) and 0.86 (synthetic data).

## 5.4 Predictor importance

The findings provided quantitative indications that the results of variable importance assessments obtained by multiple variable statistical landslide susceptibility models are not only related to underlying geomorphic processes, but also to positional errors inherent in the underlying landslide inventories. The general ranking of predictors within the models generated with real data

changed first for a mean positional accuracy of 10 m (CV) respectively 5 m (SCV) (Fig. 6A,C). In contrast, the ranking of predictors remained unchanged for the models generated with synthetic data sets up to a mean positional error of 120 m (Fig. 6B,D).

Slope had the greatest variable importance in all situations, revealing that this environmental factor played the central role to predict landslide susceptibility within each model. However, an interpretation of AUROC decreases (Fig. 6) also suggested

that the dominance of this predictor diminished as positional error increased (e.g. from 0.25 to 0.08 in Fig. 6A). An opposite, but less pronounced tendency was observed for the predictor lithology (e.g. from 0.01 to 0.05 in Fig. 6A). The importance of





the predictors slope (CV: 0.08; SCV: 0.07) and lithology (CV: 0.05; SCV: 0.06) was similar whenever the models were generated with the most inaccurate inventory and with real data.

## 6. Discussion

In general, our results were in agreement with studies stating that inventory-based errors modify the results of a landslide susceptibility analysis (Ardizzone et al., 2002; Fressard et al., 2014; Galli et al., 2008). We observed that an increasing positional error of landslide locations was generally related to an increasing distortion of subsequent modelling results. Thus, this study provides quantitative evidence for the statement that the explanatory power of a landslide susceptibility analysis increases with an increasing quality of the underlying landslide data set (Blahut et al., 2010; Fressard et al., 2014; Galli et al., 2008; Guzzetti et al., 2006; Harp et al., 2011; Petschko et al., 2015; Steger et al., 2016; van Westen et al., 2008). However, the systematic comparisons also suggested that interdependencies between inventory-based errors and subsequent modelling and validation results are complex. Thus, an identification of a generally valid threshold (i.e. the positional error should not exceed x meters), which separates acceptable models from unacceptable ones, was considered to be inappropriate. The findings provided valuable indications that the propagation of inventory-based errors is not only determined by the degree of positional inaccuracy inherent in a landslide data set, but also by a combination of additional factors. Those aspects include (i) the spatial representation of landslides and the environment (cf. Sect. 6.2), (ii) landslide sizes (cf. Sect. 6.2), (iii) the characteristics of the study area (cf. Sect. 6.3), (iv) the selection respectively parameterization of a classification method (cf. Sect. 6.4) and (v) an interplay of predictors within multiple variable models (cf. Sect. 6.5).

### 6.1 Usefulness of an additional modelling with synthetic data

The spatial information available for this study was, as presumably every "real" data set used within regional landslide susceptibility studies, considered to be imperfect. Apart from the assumption that the present inventory was affected by a land-cover related incompleteness (Petschko et al., 2015), also the available environmental data was not expected to flawlessly represent the full spectrum of landslide predisposing factors for the area. Thus, statements on how a specific data modification (in this case: using positionally-inaccurate inventories) reduced or increased the respective model's quality should be treated with caution, especially because increasing predictive performances do not necessarily reflect an increasing quality of a landslide susceptibility model (Lobo et al., 2008; Steger et al., 2016). In this respect, synthetically generated data allowed us to define a reference model that depicted a "true" association between landslides and the environment. Subsequent modelling proved valuable to gain deeper insights into the effect of inventory-based errors on modelling results and cross-check our findings. Furthermore, observed differences between modelling results obtained from the real data sets and the synthetically generated data (e.g. differences in OR-drops; cf. Fig. 4) were helpful to move our attention to related important model-influencing aspects (e.g. characteristics of the study area; cf. Sect. 6.3).





## 6.2 The spatial representation of input data and landslides size

The available landslide point inventory represents the scarp locations of mainly smaller landslide features, which tend to exhibit a distinct steep and concave morphology in comparison to its surroundings (Petschko et al., 2015). Consequently, already small changes in the point position (e.g. 5 m) may lead to the tendency that the respective features are, in the case of high-resolution slope data (e.g. 1 m x 1 m), displaced into their usually flatter vicinities. However, those immediate neighbourhoods may be represented by identical lithologies, land cover classes or soil types due to a typically coarser spatial resolution of the underlying data sets (Cascini, 2008; van Westen et al., 2008).

We argue that the applied modelling resolution of input data might not only affect the relative importance of predictors within a landslide susceptibility model (Catani et al., 2013), but also how spatial inventory-based errors are propagated into the final modelling results. For instance, the resampling of the present DTM from 1 m x 1m to the modelling resolution of 10 m x 10 m was related to an enlargement of the area covered by one grid cell (i.e. 1 m² to 100 m²) and a generalization of the topography. This combination increased the chances that nearby observations (e.g. 5 m distance from original location) were represented by an identical grid cell or cells with similar values (e.g. of slope) to the respective original location. Therefore, slope angles of landslide locations (real data) derived from a 10 m-raster were on average just 0.2° lower for a mean inventory-based inaccuracy of 5 m compared to the unmodified inventory. These similarities were expected to mainly cause the observed effect that an average inventory-based error of 5 m did almost not affect modelled relationships (Fig. 4). Thus, we argue that a resampling of predictors to a coarser resolution might not only help to reduce the impact of local topographic variations related to post-failure conditions (conditions after landslide occurrence) or data noisiness (Petschko et al., 2014a; Van Den Eeckhaut et al., 2006; van Westen et al., 2008), but also to reduce the impact of smaller inventory-based positional errors on the results of a grid-based statistical landslide susceptibility model.

A very high mean positional error of an inventory (e.g. 120 m) may require a correspondingly higher generalization of the topography to enhance the chances that the respective landslide observation is still represented by an identical respectively similar grid cell. However, those high simplifications may as well lead to the effect that actual important landslide-influencing morphological features disappear within the derivatives of a strongly resampled terrain model.

The chances that an inaccurately mapped landslide-point may still be located within the boundaries of a corresponding landslide may as well increase with an increasing size of the respective landslides. However, statistically predicting landslide susceptibility for mainly larger events may nonetheless require a substantial generalization of a grid-based topography, because the respective pre-failure conditions may considerably differ from the conditions after landslide occurrence (Hussin et al., 2016; Süzen Lüfti and Doyuran, 2004; Van Den Eeckhaut et al., 2006). Again, corresponding simplifications (i.e. resampling) might result in topographical predictors of little practical use. Such discrepancies (i.e. between pre-failure and post-failure topographies) may be counteracted for accurately mapped inventories by approximating the terrain before landslide occurrence (e.g. Van Den Eeckhaut et al., 2006). However, such approaches are likely to be inapplicable whenever only positionally erroneous inventories are available.



We finally recommend that an expected small to moderate positional inaccuracy inherent in a landslide inventory (e.g. 5 to 20 m) might partly be counteracted by resampling the respective grid-based data sets to a spatial resolution higher than the expected mean positional inaccuracy. An alternative spatial representation of the environment (e.g. terrain units, first-order catchments) (Alvioli et al., 2016; Bell et al., 2014; Guzzetti et al., 1999) might be most expedient whenever modelling with

highly inaccurate landslide inventories or erroneous inventories that mainly consist of large events is envisaged. The associated high generalization of environmental conditions within terrain-unit based models might be one reason for earlier observations that susceptibility maps generated from inventories with considerable positional disagreements appeared and performed similarly (Ardizzone et al., 2002). In this respect, recent developments which allow an automatic, but controllable (e.g. sizes of units) subdivisions of the terrain into slope units (Alvioli et al., 2016) might prove highly useful. It is, however, important

to ensure that the respective landslides are still located within the unit where they were initiated (Bell et al., 2014). Which approach to apply also depends on the ultimate aim of the final susceptibility maps, e.g. either to be used in spatial planning strategies or as part of regional landslide early warning systems.

### 6.3 The characteristics of the study area

For the real data, the modelled relationship between landslide occurrence and topographic characteristics weakened for

positional inventory errors > 10 m (Fig. 4A,C). This was attributed to the effect of positional error reducing observed topographic differences between landslide and non-landslide locations (e.g. slope angles in Fig. 2B). In the more strongly generalized synthetic area, in contrast, such changes in modelled relationships were hardly noticeable up to a mean positional error of 50 m (3D view in Fig. 3A; Fig. 4D). The differences between the synthetic data and the real data were interpreted as evidence that the geomorphic characteristics of a study area plays as well a major role on how inventory-based inaccuracies

may be propagated into modelling results. We inferred that as long as the resolution of the data is generalized to mainly present pre-failure conditions, grid-based models generated for undulating and smooth landscapes (e.g. Flysch Zone of Lower Austria) might tend to be less affected by smaller positional errors of the inventory compared to sites featuring sharp topographic transitions (e.g. terraced landscapes, cuesta landscapes, limestone areas). Based on the results we inferred that the influence of an identical inventory-based mean positional error (e.g. 20 m) on subsequent modelling results may be considerably higher for

areas exhibiting a topographically more complex terrain. This complexity is on one hand dependent on how a study site is represented (cf. Sect. 6.2), but also on the characteristics of the study area itself.

### 6.4 An additional argument on why model overfitting should be avoided

An aspect worth further consideration is whether different classification techniques are to a different extent affected by positional inaccuracies of landslide inventories. In particular, non-linear statistical models (e.g. Generalized additive models,

multivariate adaptive regression splines) or flexible machine learning techniques (e.g. Support Vector Machines, Decision Trees, Random Forest) are increasingly used because of possible gains in predictive performance (Ballabio and Sterlacchini, 2012; Catani et al., 2013; Felicísimo et al., 2013), but they bear a higher risk of overfitting to training data (Brenning, 2005,





2012a; Goetz et al., 2015a; Steger et al., 2016). In analogy to a generalized representation of the study area (cf. Sect. 6.2), we expect that a higher degree of model generalization, as provided by the less flexible techniques, may provide a certain level of protection against learning, i.e. overfitting to, errors originating from a landslide inventory.

Thus, this study supports the statement that an application of strongly generalizing classifiers might still be valuable in the

context of landslide susceptibility modelling (Brenning, 2012a; Goetz et al., 2015a; Steger et al., 2016) while we suspect that strongly overfitting classifiers are likely to reproduce inventory-based errors. This assumption is supported by earlier research, which showed that logistic regression-based landslide susceptibility models generated with rather different landslide inventories produced similar predictive performances and landslide susceptibility maps (Zêzere et al., 2009). Furthermore, Ardizzone et al. (2002) concluded that statistical models may as well substantially minimize the effect of data errors on

landslide susceptibility models.

### 6.5 Considerations of an interplay of predictors within multiple variable models

Logistic regression models, like other multiple variable classifiers, try to maximize the probability of obtaining the observations (landslide presence or landslide absence) according to the data provided (Hosmer and Lemeshow, 2000). The present findings (i.e. OR and predictor importance) provided quantitative indications that the applied models counteracted the

diminishing explanatory power of topographic variations (especially slope angle) by adjusting the weights assigned to the coarser scaled predictor lithology.

For instance, the logistic regression models produced with real data and with the unmodified inventory did not strongly accentuate differences in landslide susceptibility (cf. OR in Fig. 4B) between the Flysch (7.2 landslides per km²) and the Molasse (1 landslide per km²) since the respective models accounted for the fact that the Flysch Zone is considerably steeper

(mean slope 12.3°) compared to the Molasse (5.1°). Even though landslide densities observed for those units remained nearly unchanged when simulating an inventory-based-error (i.e. Flysch 7 and Molasse 1 landslide per km² for the 120 m mean error), modelled relationships increasingly emphasized differences between those units (Fig. 4B). Thus, we inferred that the respective models were decreasingly capable to account for local topographic variations (OR of slope are approaching the threshold of 1) with an increasing positional error of the inventory and subsequently were increasingly dependent on differences describable

by other predictors (e.g. landslide densities for lithological units). Those tendencies were also discernible for the synthetic data set, ultimately due to deteriorating topographical effects (cf. Fig. 4) and dissimilar landslide densities between lithological units (cf. Sect. 5). Variable importance assessments (e.g. decreasing importance of slope and increasing importance of lithology in Fig. 6) and the appearance of the final maps (e.g. striking appearance of the Flysch in Fig. 5F) further exposed these distortions. This study underlines that besides the applied raster resolution (Catani et al., 2013), also the inventory-based

positional accuracy influences how strongly a predictor appears to discriminate landslide-presences from landslide-absences. Consequently, we argue that potential inventory-based positional errors should additionally be taken into account, when a process-based geomorphic interpretation of statistical landslide susceptibility models is envisaged (Brenning et al., 2015; Vorpahl et al., 2012).



### 6.6 Misleading performance estimates

In general we found evidence that a point-based landslide inventory that is increasingly affected by positional errors approaches a distribution of complete spatial randomness (Cressie, 2015). Thus, statistically discriminating landslide locations from non-landslide locations becomes increasingly difficult since non-landslide observations are regularly represented by random spatial observations (e.g. Conoscenti et al., 2016; Goetz et al., 2015a). The observed decreasing predictive performances provided quantitative evidence for this assumption (CV and SCV in Fig. 5). The tendency towards diminishing differences between landslide presences and absences were also reflected by conditional frequencies of slope angles (Fig. 2B; Fig. 3G) and OR of topographic parameters (Fig. 4).

The spatial pattern of landslide susceptibility maps (e.g. cutouts in Fig. 5) visually highlighted that all models established a positive relationship between landslide occurrence and slope angle (ORs > 1.9 in Fig. 4A,D) while the lithological units Flysch (real data) respectively the unit A (synthetic data) were consistently predicted as most susceptible (Fig. 4B,E). Thus, all models predicted that the steepest parts of the Flysch Zone (respectively unit A) are highly prone to landsliding, even though the underlying landslide-presences (white dots in Fig. 5) were increasingly located on considerably flatter slopes (e.g. Fig. 2B; Fig. 3G). In analogy, the flattest parts covered by Quaternary sediments (respectively unit C) were consistently predicted as the most stable locations. From this perspective, it might become clear that models generated with erroneous inventories performed well (all AUROCs > 0.84) in predicting the original landslide position (cf. "0" in Fig. 5). This observation is consistent with our suggestion that strongly generalizing classifiers might also reduce the effect of inventory-based errors on modelling results (cf. Sect. 6.4; Ardizzone et al., 2002). However, validation results that were based on a landslide subsample of the data set used to generate the model (as within most studies) revealed apparently lower predictive performances (CV, SCV in Fig. 5). Thus, this study provides another example that misleading performance estimates may follow whenever a landslide susceptibility model is generated and validated with erroneous landslide inventories (Steger et al., 2016).

This study further underlined that the predictive performance of a statistical landslide susceptibility model is just one of many indicators of the quality and reliability of a spatial prediction model (Fressard et al., 2014; Guzzetti et al., 2006; Lobo et al., 2008; Petschko et al., 2014a; Rossi et al., 2010; Steger et al., 2016). Thus, we consider a differentiated evaluation of input data qualities and modelling results (i.e. quantitative and expert-based) as an necessary supplement to get insights into limitations and the reliability of subsequent modelling results (Demoulin and Chung, 2007; Fressard et al., 2014; Guzzetti et al., 2006; Petschko et al., 2014a; Steger et al., 2016).

### 7. Conclusion and final recommendations

There is no doubt that the explanatory power of a landslide susceptibility analysis increases with an increasing quality of the underlying landslide inventory (Fressard et al., 2014; Galli et al., 2008; Guzzetti et al., 2006; Petschko et al., 2015; Steger et al., 2016; van Westen et al., 2008). The present findings highlighted that positionally erroneous landslide inventories affected modelled relationships, variable importance assessments and the explanatory power of conventional predictive performance



estimates of a statistical landslide susceptibility model. We found valuable evidence that the impact of inventory-based positional errors on modelling results is not only dependent on the degree of inaccuracy inherent in the respective landslide-presence data, but also on an interplay of other model-influencing aspects: Besides the size (i.e. small vs. large landslides) and spatial representation of landslides (i.e. landslide scarp vs. landslide body), also the spatial representation of the environment

(i.e. raster resolution, terrain unit) was expected to determine how a specific positional error was propagated into the final results. Furthermore, also the morphological characteristics of the study area (i.e. undulating landscape vs. landscape with sharp transitions) and the applied classification method (i.e. strongly generalizing vs. highly flexible classifiers) are believed to control the extent of potentially misleading modelling results.

The present findings indicate that the propagation of inventory-based positional inaccuracies into modelling results can be

reduced by selecting an appropriate study design. Thus, we advise to accept inventory-based errors as unavoidable and then seek to obtain in-depth information on the potential limitations of present data sets. A subsequent improvement of the respective data set (e.g. updated mapping) should be the first step. For the likely case that modelling with positionally imperfect inventories cannot be avoided (e.g. Bell et al., 2014; Eeckhaut et al., 2011), we recommend to generalize input data to a coarser scale (e.g. resampling of predictors or using an alternative representation of the environment; cf. Sect. 6.2) while opting for

modelling techniques that simplify observed associations (e.g. logistic regression; cf. Sect. 6.4). The potential presence of inventory-based errors should always be taken into account since subsequent performance estimates (cf. Sect. 6.6), modelled relationships and variable importance (cf. Sect. 6.5) are likely to be distorted whenever the underlying models were generated with positionally erroneous landslide inventories.

### Acknowledgment

This article was supported by the Open Access Publishing Fund of the University of Vienna. The authors also thank several Departments of the Provincial Government of Lower Austria for providing basic data: the Geological Survey, the Dept. of Spatial Planning and Regional Policy and the Dept. of Hydrology and Geoinformation.

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

**Figure 1. Location of the study area within Austria (A), shaded relief image of the area with overlaid lithological units and spatial distribution of landslide scarps (B). The shaded relief images (top right) show typical landslides of the study area and the corresponding mapping location. Slope angles are given in C while slope orientation is represented as northness (D) and eastness (E).**



**Figure 2. Overview of the simulated positional inaccuracy of the inventory for the study area (real data). The excerpt in A shows an example of an original mapping position of a landslide scarp (red dot) and corresponding simulated positional inaccuracies (the white grid corresponds to the modelling resolution of 10 m). The conditional frequency plot in B shows that a decreasing positional accuracy of the inventory led to the effect that landslides were more likely located on flatter slopes. The histograms in C (bin width 10 m) depict the frequency of landslides (n = 591) in relation to the distance from the original mapping position for the respective simulated positional inaccuracy.**





**Figure 3. Spatial representation of the synthetically generated data set. Topographic parameters (A,B,C); Lithological and land cover units (D,E) with excerpts depicting the generation of this data sets (1: initial random raster; 2: intersection with a random raster produced at a higher spatial resolution; 3: moving window-based smoothing; 4: introduction of a slope-dependency for land cover). Landslide susceptibility according the predefined relationship and subsequent landslide sample further used as landslide presences for model realizations (F). The plot in G depicts the conditional frequency of landslide occurrence on slope angle for the respective inventories. The histograms in H (bin width 10 m) show the frequency of landslides (n = 2000) in relation to the distance from the original mapping position.**





**Table 1. Prescribed relationship between the predictors and the response variable for the synthetic data set expressed as odds ratios (OR) (first row). The model estimate (bottom row) was based on the synthetically generated landslide sample (cf. Fig. 3F) and a logistic regression model trained with the predictors slope, northness, eastness, lithology, land cover. OR for the predictor slope relate to a slope angle increase of 10°, OR for both components of slope aspect (northness, eastness) to an orientation change of 45°.**

| | Slope | Northness | Eastness | Litho. A | Litho. B | Litho. C | Pastures | Forests | Arable land |
|---|---|---|---|---|---|---|---|---|---|
| OR (prescribed) | 6 | 0.8 | 1.4 | 1 | 1 | 0.5 | 1 | 0.5 | 1 |
| OR (reference model) | 6.22 *5.4 / 7.2 | 0.76 *0.7 / 0.8 | 1.37 *1.3 / 1.5 | 1 ref. class | 0.86 *0.7 / 1 | 0.45 *0.4 / 0.6 | 1 ref. class | 0.51 *0.4 / 0.6 | 0.86 *0.7 / 1.1 |

* 95% confidence interval

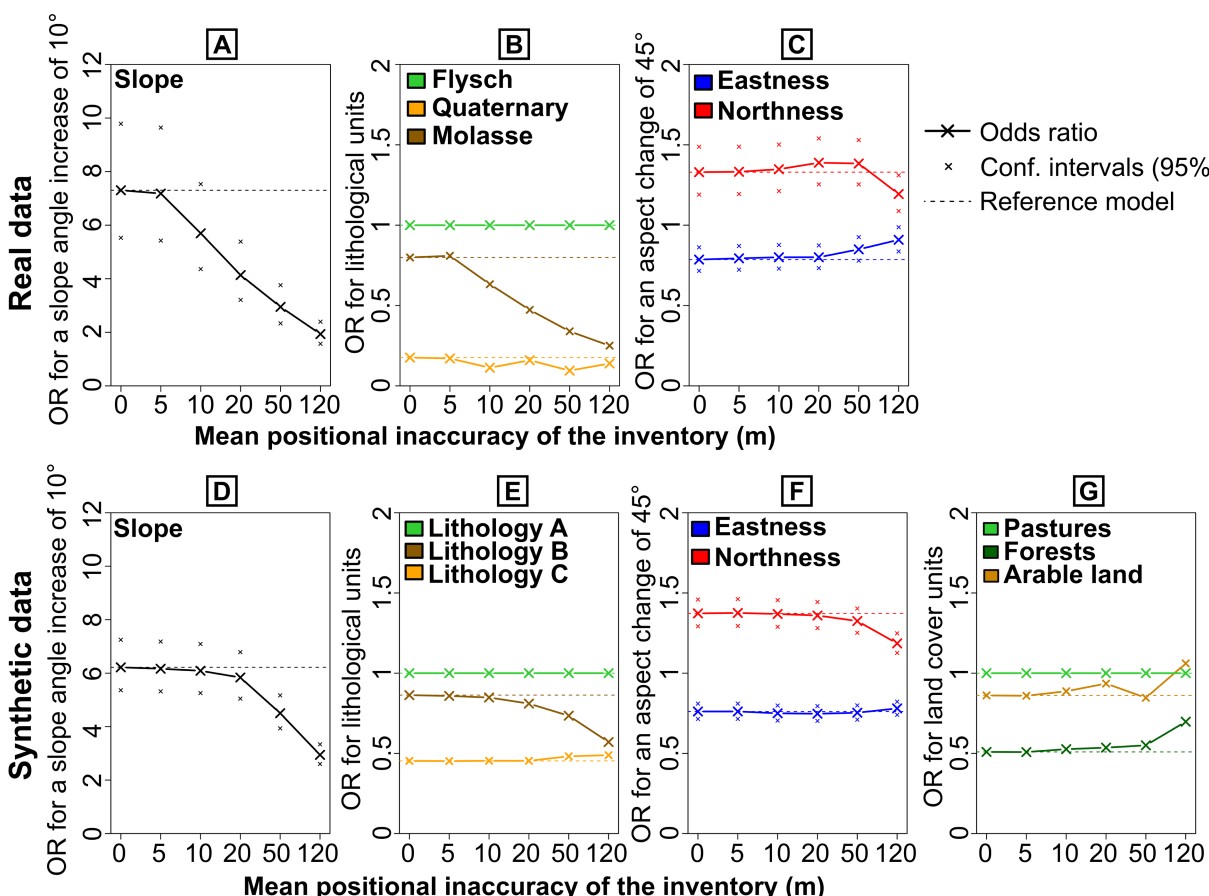

**Figure 4. Estimated odds ratios (OR) for the models generated with differently accurate landslide inventories. ORs obtained for the predictors slope (A,D), northness (C,F), eastness (C,F) and land cover (G) suggested a decreasing estimated effect of topographic and land cover variations on modelling results with an increasing positional error of the inventory (i.e. ORs are approaching the no-effect threshold value 1). ORs of the lithology classes revealed decreasing modelled chances that the Molasse (B) respectively the unit B (E) may be affected by future landsliding.**




**Figure 5. Landslide susceptibility maps (for the location of the cutouts refer to the red boxes) of models generated with differently accurate inventories (rows) for the real data (left column) and the synthetic data (right column). Corresponding predictive performances (median AUROC for CV and SCV) and the quantitative comparison with the original landslide positions ("0") are given in the respective white boxes. The cutouts additionally show the location of the positionally erroneous inventories (white dots) used as input for the respective models and the position of the original inventory (red dot). Note that maps generated with the most inaccurate inventory (F,L) are characterized by a locally more uniform spatial pattern.**




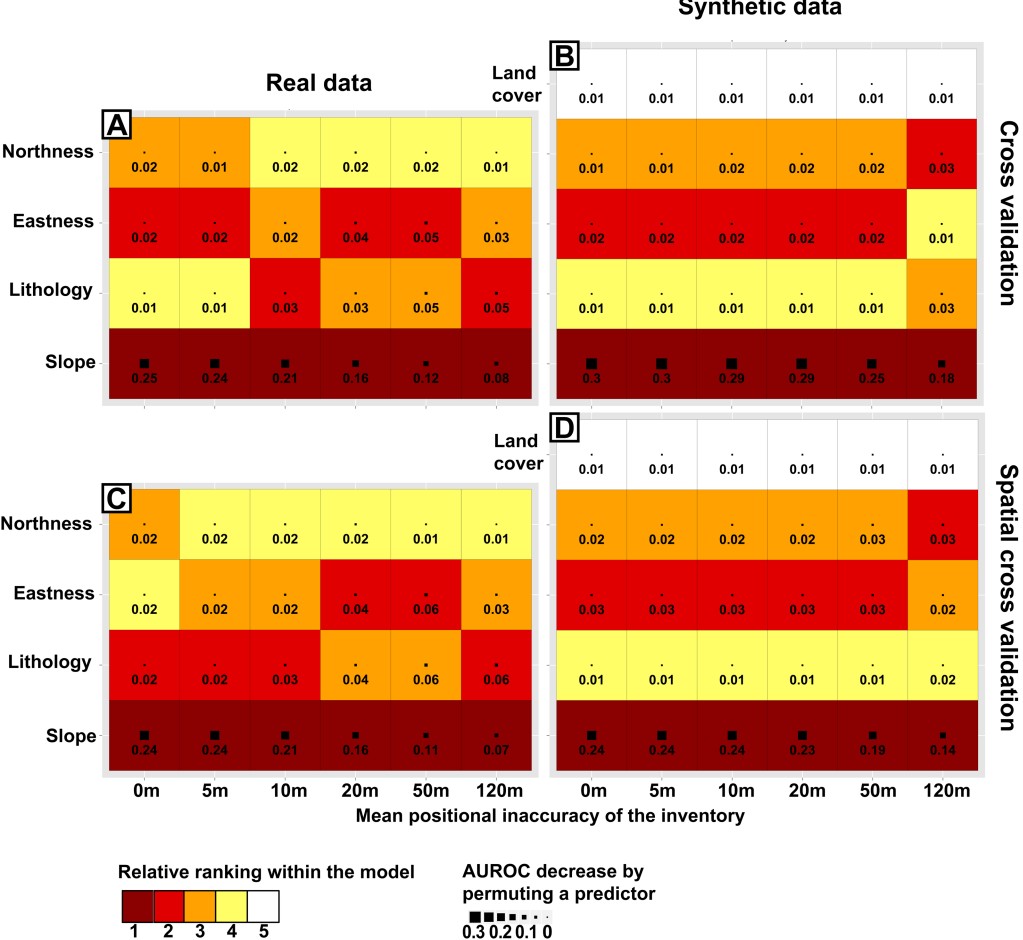

**Figure 6. Permutation-based variable importance for the respective reference models (0 m) and models generated with positionally erroneous inventories (5 m – 120 m). The relative ranking indicates that slope was the most influential predictor for each model while its estimated importance constantly decreased with an increasing inventory-based positional error (cf. AUROC decrease). The estimated importance of lithology slightly increases with a growing inventory-based positional inaccuracy.**