# Peer review of "THE PROPAGATION OF INVENTORY-BASED POSITIONAL ERRORS INTO STATISTICAL LANDSLIDE SUSCEPTIBILITY MODELS"

_Natural Hazards and Earth System Sciences, 2016_

## Referee Comment (RC1) · Anonymous Referee #1 · 30 Sep 2016

The paper deals with a very interesting topic: the propagation of errors due to an incorrect positioning of landslides into landslide susceptibility models. The issue is very important whenever a landslide susceptibility model is included in an urban or regional land planning. Moreover the effort spent in the evaluation of how the positioning error of landslides propagates into a model, contributes to understand how important is establishing standard in the preparation of landslide inventories. The paper is well organized; the figures are accurate such as the figure captions. See very few corrections in the annexed file (nhess-2016-301_Reviewer.pdf).

Please also note the supplement to this comment:
http://www.nat-hazards-earth-syst-sci-discuss.net/nhess-2016-301/nhess-2016-301-RC1-supplement.pdf

[Figure]

[Figure]

**Supplement:**

[revised manuscript text omitted]

---

## Referee Comment (RC2) · Anonymous Referee #2 · 18 Oct 2016

Review of the manuscript "The propagation of inventory-based positional errors into statistical landslide susceptibility models" By Stefan Steger, Alexander Brenning, Rainer Bell, and Thomas Glade

General Comments The manuscript of Steger and co-authors entitled "The propagation of inventory-based positional errors into statistical landslide susceptibility models" is an interesting well-structured and well-written manuscript that addresses relevant scientific and technical questions which are within the scope of NHESS. However it needs moderate revisions prior to be published.

Specific Comments

1 - Although it is reasonable to model landslide susceptibility using points (centroids), namely when landslides are small in size, in the opinion of the reviewer, the validation of

predictive models should be made with landslide areas (polygons), because a landslide is not a point.

2 - Additional information should be given regarding landslide inventory and sources used to construct it, namely: a) Dates of airborne laser scanning and ortophotos used for landslide inventorying; b) What is the typical depth of the shallow landslides? c) Do authors have any idea about the permanence time of landslides in the landscape? What is the possible range of age of inventoried landslides? d) What is the scale of the digital geological map used to derive lithological units?

3 - The landslide susceptibility was assessed using only 4 predictor themes: slope, lithology and aspect (two themes). In the opinion of the reviewer this is too restrict. The authors used a test site with 100 km2 where 591 landslides were inventoried as point. These landslides are mostly concentrated over the Flysch lithologic unit that spreads over the major part of the study area (81%). So, in this case, in the opinion of the reviewer, the lithological layer is not a good theme to discriminate between stable and unstable areas. This should be discussed in the manuscript. In addition, the spatial relationship between slope aspect and landslide distribution seems to be weak (i.e. the aspect is not very sensitive to positional errors of landslide points), which turns the landslide susceptibility mostly dependent on the variable Slope. Why did not you use other variables extracted from the DTM like slope curvature or the Slope over Area Ratio?

4 - Regarding the predictive performance, it is not clear which data was used to validate the predictive models. Apparently, the points with errors were used to validate, but this should be clearly stated.

5 - In section 4.3 authors state that "An expert-based evaluation of the final results was conducted by comparing all modelled relationships and maps with the results of those models that were assumed to be less affected..." In the opinion of the reviewer, this statement is not enough clear and needs to be better described.

[Figure]

6 - Authors developed "synthetic data" that is a virtual terrain, whose construction and justification needs to be improved. In particular, it should be explained the reason to use the land cover for the synthetic data while this theme was refused for the model with "real data". Furthermore, authors state that "A spatially balanced landslide data set was generated according to Theobald et al. (2007)". Please, provide more information about this procedure. Also, authors generate a sample containing 2000 "landslide points" which apparently is a very large number when compared with the 591 landslides inventoried in an area equivalent in size with the "virtual study area".

Technical corrections Page 11 – line 16 "(iv) the selection respectively parameterization of a classification method" Something is missing in this peace of text.

Page 17 – line 24 Brenning, A.: Spatial cross-validation and bootstrap for the assessment of prediction rules in remote sensing: The R package sperrorest, in Geoscience and Remote Sensing Symposium (IGARSS), 2012 IEEE International, pp. 5372–5375. Available from: http://ieeexplore.ieee.org/xpls/abs_all.jsp?arnumber=6352393 (last access 22 April 2016), 2012. 2012b instead of 2012

Page 19 – line 4 Petschko, H., Bell, R. and Glade, T.: Relative Age Estimation at Landslide Mapping on LiDAR Derivatives: Revealing the Applicability of Land Cover Data in Statistical Susceptibility Modelling, in Landslide Science for a Safer Geoenvironment, edited by K. Sassa, P. Canuti, and Y. Yin, pp. 337–343, Springer International Publishing. Available from: http://link.springer.com/chapter/10.1007/978-3-319-05050-8_53 (last access 26 July 2016), 2014. 2014b instead of 2014

Page 19 This reference is not used in the text? Tien Bui, D., Pradhan, B., Lofman, O. and Revhaug, I.: Landslide Susceptibility Assessment in Vietnam Using Support Vector Machines, Decision Tree, and Naive Bayes Models, Mathematical Problems in Engineering, 2012, e974638, 45 doi:10.1155/2012/974638, 2012

Figure 5 The color palette is not easy to distinguish landslide susceptibility in the maps. Please, use a more contrasting color palette.

[Figure]

Figure 6 Indicate what is A,B,C and D in figure caption.

[Figure]

---

## Author Comment (AC1) · 10 Nov 2016

We thank the anonymous referee #1 for the constructive comments on the present manuscript. We were very pleased that the topic tackled by our study was judged as "very interesting" and "very important whenever a landslide susceptibility model is included in an urban or regional land planning" and that the paper was considered to be "well organized" with accurate figures. We were also delighted that only "very few corrections" were considered as necessary prior to publication. For details, please follow our point-by-point replies given in the supplement (as well as the revised figures).

Please also note the supplement to this comment:
http://www.nat-hazards-earth-syst-sci-discuss.net/nhess-2016-301/nhess-2016-301-AC1-supplement.pdf

[Figure]

[Figure]

**(a)**

N

150 km

Lower Austria

**Study area**

30 m

30 m

• Landslides (n = 591)

**Slope**
> 30°

0°

**(b)**

2 km

**Lithology**

Quaternary

Molasse

Flysch

**(c)**

**Aspect**
1 (d: North; e: East)

-1 (d: South; e: West)

3 km

**(d)**

**(e)**

**Fig. 1.**

[Figure]

**Landslide inventory used as a response variable:**

- Unmodified ("0")
- Inaccurate ("μ5")
- Inaccurate ("μ10")
- Inaccurate ("μ20")
- Inaccurate ("μ50")
- Inaccurate ("μ120")

**Fig. 2.**

**Slope**
> 30°
0°

**Aspect**
1 (b: North; c: East)
-1 (b: South; c: West)

**Lithology**
A
B
C

**Land cover**
Pastures
Forests
Arable land

**Probability raster**
1st quartile [< 0.0006]
2nd quartile [0.0006 - 0.0013]
3rd quartile [> 0.0013 - 0.0033]
4th quartile [> 0.0033]
Landslide inventory (n = 2000)

**Fig. 3.**

[Figure]

**Fig. 4.**

**Real data**

**Synthetic data**

Mean positional inaccuracy of the inventory (m)

0 m: Unmodified
(a)
AUROC
- CV: 0.845
- SCV: 0.829
- "0": 0.849

(g)
AUROC
- CV: 0.864
- SCV: 0.864
- "0": 0.865

5 m
(b)
AUROC
- CV: 0.841
- SCV: 0.825
- "0": 0.849

(h)
AUROC
- CV: 0.863
- SCV: 0.863
- "0": 0.865

10 m
(c)
AUROC
- CV: 0.826
- SCV: 0.808
- "0": 0.848

(i)
AUROC
- CV: 0.862
- SCV: 0.862
- "0": 0.865

20 m
(d)
AUROC
- CV: 0.793
- SCV: 0.773
- "0": 0.844

(j)
AUROC
- CV: 0.856
- SCV: 0.856
- "0": 0.865

50 m
(e)
AUROC
- CV: 0.764
- SCV: 0.743
- "0": 0.840

(k)
AUROC
- CV: 0.83
- SCV: 0.83
- "0": 0.864

120 m
(f)
AUROC
- CV: 0.686
- SCV: 0.653
- "0": 0.843

(l)
AUROC
- CV: 0.753
- SCV: 0.746
- "0": 0.859

**Landslide susceptibility**

1    0

**Inventory**

- • Unmodified inventory ("0")
- ○ Positional inaccurate inventory

**Fig. 5.**

**Real data**

**Synthetic data**

**(a)**

**(b)**

Cross validation

**(c)**

**(d)**

Spatial cross validation

Mean positional inaccuracy of the inventory

Relative ranking within the model

1  2  3  4  5

Median AUROC decrease by
repeatedly permuting a predictor

0.3 0.2 0.1 0

**Fig. 6.**

**Supplement:**

We thank the anonymous referee #1 for the constructive comments on the present manuscript. We were very pleased that the topic tackled by our study was judged as *"very interesting"* and *"very important whenever a landslide susceptibility model is included in an urban or regional land planning"* and that the paper was considered to be *"well organized"* with accurate figures. We were also delighted that only *"very few corrections"* were considered as necessary prior to publication. For details, please follow our point-by-point replies:

- **RC1:** Original comment of referee #1
- **AR:** Response of the authors (black) and changes/changed text segments (blue)
- Page and line numbers refer to the revised version of the manuscript (e.g. p.2 line 8)

**RC1:** No abbreviation, please.
**AR:** Corrected. We dropped this bracket "(n = 591)" (p.1 line 15), because this information is provided within Sect. 3.1 (p.4 line 21) and within Figure 1.

**RC1:** "."
**AR:** Done (p.2 line 4).

**RC1:** Replace Fig. 1 with Fig. 1B
**AR:** We changed the arrangement of the figures (i.e. slope map above lithological map) as suggested by the referee to allow a consequential description of the maps. The color ramp of the slope map was adapted (i.e. ramp without red) to ensure visibility of the landslide points (red). The color ramp of the slope map shown in Fig. 3a was as well changed in order to ensure consistency with Fig. 1. The abbreviation "Fig. 1" was replaced with "Fig. 1b" as suggested.

**RC1:** Replace Fig. 1B with Fig. 1C
**AR:** Done. Changed to Fig. 1c.

**RC1:** Replace "Fig. 1B" with "Fig. 1C" and "Fig. 1C" with "Fig. 1B"
**AR:** Done. Changed to Fig. 1c and Fig. 1b respectively.

**RC1:** Which type of analysis? Could you add a reference here?
**AR:** Thank you for the comment. We extended this paragraph to clarify our approach. Furthermore, we included an additional reference (p.5 line 31, p.6 line 1f):
*"Within this analysis, 65 of the 681 analyzed landslide database entries of the Building Ground Registry (more information refer to Schwenk, 1992 and Steger et al., 2016) contained a quantitative estimate on the positional accuracy of the respective landslide point location. The derived mean positional error of 120 m (standard deviation of 84 m) was directly adopted to specify the inaccuracy of the most erroneous inventory."*

**RC1:** "pixilated" (check, please)
**AR:** We changed this expression to *"noisy appearing grid files"*. (p.8 line 29)

**RC1:** Labels of panels must be included with brackets around letters being lower case: (a)
**AR:** Corrected for all figures. All cross-references in the text and figure captions were adapted accordingly.

---

## Author Comment (AC2) · 10 Nov 2016

We cordially thank the anonymous referee #2 for the detailed and constructive comments on our paper. We are delighted that our paper was perceived as "an interesting well-structured and well-written manuscript that addresses relevant scientific and technical questions which are within the scope of NHESS". We followed the referee's suggestions ("moderate revisions prior to be published") and revised the paper accordingly. For details, please follow our point-by-point replies (supplement).

Please also note the supplement to this comment:
http://www.nat-hazards-earth-syst-sci-discuss.net/nhess-2016-301/nhess-2016-301-AC2-supplement.pdf

[Figure]

[Figure]

**Supplement:**

We cordially thank the anonymous referee #2 for the detailed and constructive comments on our paper. We are delighted that our paper was perceived as *"an interesting well-structured and well-written manuscript that addresses relevant scientific and technical questions which are within the scope of NHESS"*. We followed the referee's suggestions (*"moderate revisions prior to be published"*) and revised the paper accordingly. For details, please follow our point-by-point replies:

- **RC2:** Original comment of referee #2
- **AR:** Response of the authors (black) and changes/changed text segments (blue)
- Page and line numbers refer to the revised version of the manuscript (e.g. p.2 line 8)

**1 RC2:** Although it is reasonable to model landslide susceptibility using points (centroids), namely when landslides are small in size, in the opinion of the reviewer, the validation of predictive models should be made with landslide areas (polygons), because a landslide is not a point.

**1 AR:** The referee addresses a very important issue within the field of statistical landslide susceptibility modelling, namely the sampling of landslides observations. It was pointed out by the referee (as well as by the already cited studies Atkinson et al., 1998; Goetz et al., 2015a; Petschko et al., 2014a; Van Den Eeckhaut et al., 2006 cf. p.4 line 18f) that statistical landslide susceptibility models are regularly constructed by using one point per landslide observation (i.e. preferably in the landslide initiation area), even though landslides are spatial phenomena with a certain spatial extent. The expected median landslide sizes of the study area are described in p.4 line 6f (787 m² (Flysch); 1,189 m² (Molasse); 1,260 m² (Quaternary)).

In this context, we want to emphasize that the points used within this study solely refer to landslide scarp locations. Thus, the respective landslide susceptibility maps solely provide information on where landslides of the slide-type movement are more likely to be initiated. To make this explicit, we added the following sentence to p.4 line 19f: "*Since the respective points stand representative for the main scarp location (cf. Petschko et al., 2016), subsequent landslide susceptibility maps provide an estimate on where landslides are more likely to be initiated.*"

In case the referee's comment referred to polygons that represent the area of landslide initiation, we want to explain why we believe that the decision to model and validate with one point per landslide observation is straightforward in the context of statistical landslide susceptibility modelling: Using one point per observation guarantees an equal treatment of small and large landslides. For validation, usage of a polygon (represented by multiple pixels or points) is expected to introduce the undesired effect that the respective validation results would heavily rely on a small number of large landslides. We believe that weighting for size should be avoided when modelling and validating the models in order to be consistent with the landslide susceptibility definition, which formally excludes landslide magnitude. To clarify, we rephrased the definition given in p.2 line 6f which now reads as follow: "*The term landslide susceptibility refers to the likelihood of a certain location to be affected by upcoming landslides without taking into account the potential temporal occurrence or magnitude of landslide events (Brabb, 1984; Corominas et al., 2013; Fell et al., 2008; Guzzetti et al., 1999, 2005).*"

We also want to emphasize that we are aware that the present point-based landslide information is not perfect (as presumably every historic landslide inventory). Therefore, we highlighted that the results obtained by the "real data set" should be interpreted with caution (p.8 line 14f): "*especially due to the unavailability of perfect and accurate spatial information (i.e. landslide data and predictors) and the inherent subjectivity involved during model construction (e.g. predictor and classifier selection).*" This was one major reason why we decided to validate our results with "unbiased" synthetic data sets (cf. 4.6 Generation of synthetic data; 6.1 Usefulness of an additional modelling with synthetic data).

**2 RC2:** *Additional information should be given regarding landslide inventory and sources used to construct it, namely:*
*a) Dates of airborne laser scanning and ortophotos used for landslide inventorying;*

**2a AR:** We added this information (p.4 line 15f): "*[…] of a 1 m x 1 m airborne laser scanning (ALS) digital terrain model (DTM; flight campaign: 2006-2009), supported by interpreting two orthophotos (flight campaigns: 2000-2004 and 2007-2008)."* To ensure conciseness of the paper and to avoid

repetitions with earlier studies (i.e. Petschko et al., 2016), we opted to keep the inventory chapter (3.1) rather compact (180 words). However, the final sentence of this section points to the previously mentioned study, which thoroughly describes the generation of the inventory used (p.4 line 21f): *"Further information on the mapping of this inventory and its related advantages and disadvantages are described by Petschko et al. (2016)."*

**b)** *What is the typical depth of the shallow landslides?*
**2b AC:** Precise information on the depth of the landslides cannot be given, due to an absence of such information and the fact that just a part of the mapped landslides were validated in the field (cf. Petschko et al., 2016). However, we addressed landslide size, which is regularly considered as a proxy for landslide magnitude in p.4 line 6f: *"The majority of landslides in the area are relatively small and shallow. Petschko et al. (2016) investigated landslide sizes for the districts Amstetten, Waidhofen/Ybbs and Baden and found a median size of landslides of the slide-type movement of 787 m² for the Rheno-Danubian Flysch Zone, 1,189 m² for the Molasse Zone and 1,260 m² for Quaternary sediments."*
We now added new information on the expected landslide-depths to p.4 line 8f: "*Based on an analysis of landslide archive entries (i.e. Building Ground Registry), Bell et al. (2014a) estimated a median landslide depth of 1.7 m (mean 2.2 m) for 142 landslides recorded for the district Waidhofen/Ybbs.*"
We added this new reference to the paper:
*Bell, R., Petschko, H., Proske, H., Leopold, P., Heiss, G., Bauer, C., Goetz, J., Granica, K. and Glade T.: Methodenentwicklung zur Gefährdungsmodellierung von Massenbewegungen in Niederösterreich – MoNOE, Final project report, pp. 224, Vienna., 2014a.*
We believe that more information on landslide magnitude (or landslide depth) would not contribute substantially to the objectives of the study, especially because landslide magnitude is not accounted for in conventional statistical landslide susceptibility models (as explained before).

**c)** *Do authors have any idea about the permanence time of landslides in the landscape? What is the possible range of age of inventoried landslides?*
**2c AR:** These are very interesting questions that were previously tackled by the co-authors, also for parts of the study area (compare the already cited papers Bell et al. (2012) and Petschko et al. (2014b)). In summary, those studies conclude that a specific age of the respective landslides cannot be deduced by interpreting its geomorphic footprint, while relative age approximations should as well be treated with caution, especially because of an intensive anthropogenic impact which varies between land cover units (e.g. regular land levelling on pastures). We are convinced that the present inventory contains both, recent events (before 2009) as well as older ones (e.g. few hundred years). At the same time, we also assume that especially larger events may be overrepresented within the present inventory because smaller features are expected to be eroded more quickly and therefore not visible on recent remote sensing data. This is another reason why we believe that, in our case, modelling and validating with polygons (which may emphasize such a size-bias) should be avoided. The modified definition of landslide susceptibility should further clarify that the time component is not accounted for within this study (p.2 line 6f): "*The term landslide susceptibility refers to the likelihood of a certain location to be affected by upcoming landslides without taking into account the potential temporal occurrence or magnitude of landslide events (Brabb, 1984; Corominas et al., 2013; Fell et al., 2008; Guzzetti et al., 1999, 2005).*" However, the mentioned land cover related bias was one reason why we excluded land cover from the analysis within the real data set (for more details refer to "**8 AR**" below).

**d)** What is the scale of the digital geological map used to derive lithological units?
**2d AR:** The scale of the available digital geological map is 1:200,000 (the inserted abbreviation "GK200" stands for "Geologische Karte 1:200 000"). We made this information explicit by rephrasing the respective text passage (p.5 line 6f) to "*This information was obtained from a digital geological map of Lower Austria (GK200) available at a scale of 1:200,000 and resampled to the modelling resolution of 10 m x 10 m.*"

**3 RC2:** The landslide susceptibility was assessed using only 4 predictor themes: slope, lithology and aspect (two themes). In the opinion of the reviewer this is too restrict.

**3 AR:** The first sentence of Sect. 3.2 (predictor variables) reads as follows: *"The number and type of predictors used within statistical landslide susceptibility analyses varies greatly depending on the scope of the study, the type of the investigated landslides, the characteristics of the area and data availability (Guzzetti et al., 1999; Van Westen et al., 2008)."* As formulated in the introduction (Sect. 1), the main scope of this study was to thoroughly investigate the propagation of inventory-based positional errors into the final results from a variety of perspectives (e.g. relationship of each variable to the response, importance of each variable, applying three 'types' of quantitative validation techniques, spatial pattern of maps) in a tangible and transparent way.

Thus, we decided to keep the models simple and justified this decision in p.4 line 26f: *"To enhance traceability of modelling results, we opted to include only few but widely used predictors within the analyses. All models were generated with […]"*. We further explained our decision a second time in p.5 line 17f: *"In this study we aimed to analyze all data in a tangible and traceable way. Therefore, we avoided using less interpretable classifiers or a high number of predictors."*

We are fully aware that the decision to generate and/or select a specific predictor set is subjective and influences modeling outcomes. We now re-phrased the text as follows (p.8 line 13f): *"None of the susceptibility models generated for the present study area can be considered to reflect a true and unbiased relation between landslides and environmental conditions, especially due to the unavailability of perfect and accurate spatial information (i.e. landslide data and predictors) and the inherent subjectivity involved during model construction (e.g. predictor and classifier selection)."* In our opinion, this statement well summarizes the reasons why we decided to additionally validate all analyses made within this study with synthetic data. We also discussed later (p.12 line 19f) that the *"available environmental data was not expected to represent the full spectrum of landslide predisposing factors for the area".* With the synthetic data sets we were able to counteract this issue as described thoroughly within Sect. 4.6.

**4 RC2:** The authors used a test site with 100 km2 where 591 landslides were inventoried as point. These landslides are mostly concentrated over the Flysch lithologic unit that spreads over the major part of the study area (81%). So, in this case, in the opinion of the reviewer, the lithological layer is not a good theme to discriminate between stable and unstable areas. This should be discussed in the manuscript.

**4 AR:** We appreciate the referee's comment and point out that it is true that landslide densities differ among the lithological units (as described in Sect. 5): Flysch Zone (7.2 landslides/km²), Molasse (1), Quaternary (0.4). From a quantitative point of view, lithology might therefore be highly suitable to discriminate potential landslide prone regions from less stable areas, because the substantial variations described are well describable by the lithologic layer within a statistical model. This might especially be the case when bivariate models are applied. However, our multiple variable models showed (from our perspective) interesting tendencies that are directly related to the referee's comments. We dedicated a full discussion chapter ("6.5 Considerations of an interplay of predictors within multiple variable models") to explain this issue. In summary, this chapter highlights that the models produced with an unbiased inventory (p.15 line 14f) *"[…] did not strongly accentuate differences in landslide susceptibility (cf. OR in Fig. 4b) between the Flysch (7.2 landslides per km²) and the Molasse (1 landslide per km²) since the respective models accounted for the fact that the Flysch Zone is considerably steeper (mean slope 12.3°) compared to the Molasse (5.1°)."* Interestingly, this tendency changed as soon as the inventory-based error was high as described in p.15 line 17f: *"Even though landslide densities observed for those units remained nearly unchanged when simulating an inventory-based-error […], modelled relationships increasingly emphasized differences between those units (Fig. 4b)."* Ultimately, this is the reason why the maps (e.g. Fig. 5f) generated with the most inaccurate inventory strongly accentuated the (p.11 line 3) *"the silhouettes of the Flysch Zone (cf. Fig. 1c)."* The synthetic area was not affected by such an evident spatial distribution of lithological units, but confirmed the results previously discussed.

**5 RC:** In addition, the spatial relationship between slope aspect and landslide distribution seems to be weak (i.e. the aspect is not very sensitive to positional errors of landslide points), which turns the landslide susceptibility mostly dependent on the variable Slope. Why did not you use other variables extracted from the DTM like slope curvature or the Slope over Area Ratio?

**5 AR:** The referee is correct that the relationship between slope aspect and landslide occurrence is relatively weak in comparison to the association observed with slope angle (especially for the reference models). This is reflected by modelled relationships (i.e. OR) as well as by variable importance estimates. We re-checked the paper and found that the relationship between slope aspect and landslide occurrence was explicitly mentioned just once in the results section (Sect. 5.1)), but several times summarized by the terms "topographic variations" or "topography" (i.e. these terms were meant to refer to slope angle and slope aspect).

We now re-phrased several text segments to further clarify the relations observed between slope aspect and landslide occurrence:

- p.10 line 29f: *"Thus, a high positional error of the inventory (mean error: 120 m) was observed to result in susceptibility maps that were less influenced by local slope angle and slope aspect variations"*

- p.13 line 18f: *"A very high mean positional error […] may require a correspondingly higher generalization of the topographic variables, such as slope angle or slope aspect, to enhance the chances that the respective landslide observation is still represented by an identical respectively similar grid cell."*

However, we want to stress that we found it particularly valuable to focus our discussion on the predictors slope and lithology, because (i) those predictors showed the most distinct sensitivity to landslide inventory-based errors and (ii) revealed (from our perspective) important effects in the context of the study's objective. These effects (profoundly discussed in Sect. 6.5) are primarily related to the spatial interrelation between the mentioned predictors (i.e. steepness varies between lithological units). In summary, the results highlight that the reference models were highly dependent on slope angle, while the mentioned interrelation between lithology and slope angle caused the effect that the models generated with inaccurate inventories were much less depended on topographic variations (and more reliant on lithology).

Our previous reply to the referee (cf. 3 AR) further justifies the conducted predictor selection. We agree that using a larger set of predictors (e.g. including the mentioned variables curvature, slope over area ratio) may be especially straightforward whenever a pure prediction is envisaged. However, to ensure interpretability and an in-depth evaluation of the results (i.e. permutation-based variable importance is computational intensive), we opted for a smaller predictor set. Again, we want to point out that the synthetic data was expected to represent the full spectrum of predictors for the respective inventory and thus proved useful to cross-check our results.

**6 RC2:** Regarding the predictive performance, it is not clear which data was used to validate the predictive models. Apparently, the points with errors were used to validate, but this should be clearly stated.

**6 AR:** We thank the referee for this comment and agree that additional information is required to clarify our "three-fold" quantitative validation strategy. We thoroughly revised the respective method section (Sect. 4.4) which now reads as follows:

"*The prediction skills of the models were estimated by calculating the AUROC by applying two partitioning techniques, implemented in the R package "sperrorest" (Brenning, 2012b), namely k-fold cross validation (CV) and k-fold spatial cross validation (SCV). In contrast to single hold-out validation, CV and SCV are not based on one single split of the training and test sample (e.g. 80% for calibration and 20% for validation), but on a repeated partitioning of the original sample into k subsamples. In each iteration, a performance measure (e.g. AUROC) is estimated for one of the k subsamples, while the remaining (k-1) subsamples are combined into a training set that is used to calibrate the model. Thus, validation results that are based on CV and SCV are not dependent on one specific sample split. In fact, CV as well as SCV allow that all available data can be used to validate and to calibrate the final models. CV is based on a repeated non-spatial random splitting, whereas SCV is performed*

*spatially and consists of a repeated spatial partitioning of the training sample and test sample (Brenning, 2012b; Ruß and Brenning, 2010). In this study, the predictive performance of all models was estimated by repeating CV and SCV 50 times with 10 folds per repetition. More specifically, within each of the 50 repetitions, each observation of the response variable was applied nine times to calibrate the model and once to test the predictive performance. The presented AUROC values refer to the median of these 500 estimates. Previous studies emphasized the suitability of CV and SCV in the context of landslide susceptibility modelling (Goetz et al., 2015a; Petschko et al., 2014a; Steger et al. 2016).*
*Furthermore, an additional validation strategy was applied in order to quantitatively evaluate and compare the ability of all models to "predict" landslide presences and landslide absences of the unmodified response variable. For this purpose, the AUROC was used to compare the predictions of each model with the unmodified landslide inventory (i.e. unaffected by an artificially introduced positional error). This metric relates to the goodness of model fit, whenever the respective models were calibrated with an unmodified data set.*"

We slightly changed the legend within Figure 6 to further clarify that the respective numbers relate to repeated measurements "*Median AUROC decrease by repeatedly permuting a predictor*"

**7 RC2:** In section 4.3 authors state that "An expert-based evaluation of the final results was conducted by comparing all modelled relationships and maps with the results of those models that were assumed to be less affected: : :" In the opinion of the reviewer, this statement is not enough clear and needs to be better described.

**7 AR:** We understand that this relatively short subchapter, and probably the expression "expert-based evaluation" may confuse the reader. In fact, the final results were not judged by external experts, as the expression "expert-based" might suggest. This evaluation "solely" related to internal comparisons among the models. We revised the text and left out the expression "expert-based" (which originally should solely point out that this evaluation was not purely number-driven). The thoroughly revised subsection reads as follows (Sect. 4.3):

"*An additional evaluation of the final results was conducted by comparing all modelled relationships and maps with the results obtained by the reference models that were generated with the original (i.e. unmodified) inventory. These references were assumed to be less affected (i.e. reference model for the real data) or unaffected (i.e. reference for the synthetic data) by inventory-based positional errors. We considered a model or map to be strongly affected by an inventory-based positional inaccuracy if the modelled relationships (i.e. OR of predictors) respectively the spatial pattern of the maps differed substantially from their previously defined references. We therefore considered the respective positional error to have little effect in all cases where the estimated OR and the susceptibility maps were similar to their references.*"

**8 RC2:** Authors developed "synthetic data" that is a virtual terrain, whose construction and justification needs to be improved. In particular, it should be explained the reason to use the land cover for the synthetic data while this theme was refused for the model with "real data".

**8 AR:** As several previous replies to the referee (i.e. **1 AR, 3 AR, 4 AR, 5 AR**) highlight, using synthetic data proved highly useful to validate the results under controlled conditions. We recognized during literature review that sensitivity analyses of landslide susceptibility models (e.g. varying data qualities, varying predictors) are usually conduced solely on the basis of real world data sets, even though spatial information (i.e. predictors, inventories) available for large study areas can be considered as imperfect (as pointed out in Sect. 6.1 "Usefulness of an additional modeling with synthetic data"). We believe that our approach can thus be considered as innovative and highly useful in the context of the scope of the study. Since we recognized that a thorough justification of the construction of the synthetic data set is needed, we devoted the most extensive method chapter for explaining its generation (cf. Sect. 4.6) and discussed its usefulness within an own discussion section (cf. Sect. 6.1). Furthermore, Figure 3, its caption and the excerpts are expected to further enhance traceability. As described and shown in Fig. 3a, a topographically diverse terrain was generated by strongly generalizing (i.e. smoothing) an available DTM. We expanded our justification to smooth the

DTM (p.8 line 27f): *"[...] which was* expected to be unaffected by local data noise and *to represent undisturbed pre-failure conditions (cf. Van Den Eeckhaut et al., 2006)."*

The referee also asked why we decided to use land cover for the synthetic data, and omitted this predictor within the real data set. We are aware that land cover is regularly used within statistical landslide susceptibility models, especially to represent hydrological (e.g. evapotranspiration, interception) or geomechanical (e.g. roots cohesion) effects. Since land cover is regularly used in statistical models and we also believe that it influences slope stability in our area, we originally planned to introduce this predictor in the real data set. However, we finally decided that, in our specific case, introducing land cover may not be straightforward due to a high potential of obtaining spurious correlations. We now revised the text part which justifies our decision to be more clear (p.5 line 8f):
*"We decided to exclude land cover from the analysis for two reasons. Firstly, the available recent land cover data does not necessarily correspond to the land cover present at the time of landslide occurrence, due to constantly ongoing land use changes* and the fact that landslide age is expected to vary substantially within the present inventory *(Petschko et al., 2014b; van Westen et al., 2008). Secondly, an omission of land cover as a predictor was expected to reduce the chances that the suspected land cover related-bias (e.g. overrepresentation of landslides in forested areas, cf. Bell et al., 2012; Petschko et al., 2016) is directly propagated into the final results (cf. Steger et al., 2016)".*

Both previously mentioned reasons do not apply for the synthetic data set (this is another benefit of modeling with synthetic data). Thus, we decided to include land cover and now made this decision explicit (p.5 line 13f). "*However, land cover was introduced as a predictor within the synthetic data set, because the respective landslide data set was not defined to be affected by a systematic error, while land cover was specified to be static in time (cf. Sect. 4.6).*"

**9 RC2:** Furthermore, authors state that "A spatially balanced landslide data set was generated according to Theobald et al. (2007)". Please, provide more information about this procedure. Also, authors generate a sample containing 2000 "landslide points" which apparently is a very large number when compared with the 591 landslides inventoried in an area equivalent in size with the "virtual study area".

**9 AR:** The mentioned sampling approach developed by Theobald et al. (2007) was adopted to "spread" the 2000 landslide points across the study area on the basis of the previously generated probability raster (which stands representative for the true landslide susceptibility). We revised the respective text to further explain this approach (p.9 line 15f): "*A sampling approach developed by Theobald et al. (2007) was adopted to spatially distribute landslide initiation zones (i.e. represented by points) according to the predefined relationships. More precisely, the mentioned probability raster was used to control sampling intensity during the generation of 2000 spatially balanced landslide initiation points (i.e. raster cells with high probabilities are more likely selected as landslide location) (Theobald et al., 2007). This comparably high number of landslide points was chosen to assure a high explanatory power of the empirical results while simultaneously assuring computational feasibility.*"
Again, this paragraph (implicitly) emphasizes the usefulness of the synthetic data set (i.e. very high sample size).

**Technical corrections**

**RC:** Page 11 – line 16 "(iv) the selection respectively parameterization of a classification method" Something is missing in this peace of text.
**AR:** We agree that this formulation might confuse the reader at this stage of the paper. Changed to "*(iv) the selection of a classification method*"

**RC:** Page 17 – line 24 Brenning, A.: Spatial cross-validation and bootstrap for the assessment of prediction rules in remote sensing: The R package sperrorest, in Geoscience and Remote Sensing Symposium (IGARSS), 2012 IEEE International, pp. 5372–5375. Available from:

http://ieeexplore.ieee.org/xpls/abs_all.jsp?arnumber=6352393 (last access 22 April 2016), 2012. 2012b instead of 2012

**AR:** We thank the referee for highlighting this error. Corrected.

**RC:** Page 19 – line 4 Petschko, H., Bell, R. and Glade, T.: Relative Age Estimation at Landslide Mapping on LiDAR Derivatives: Revealing the Applicability of Land Cover Data in Statistical Susceptibility Modelling, in Landslide Science for a Safer Geoenvironment, edited by K. Sassa, P. Canuti, and Y. Yin, pp. 337–343, Springer International Publishing. Available from: http://link.springer.com/chapter/10.1007/978-3-319-05050-8_53 (last access 26 July 2016), 2014. 2014b instead of 2014

**AR:** Corrected.

**RC:** Page 19 This reference is not used in the text? Tien Bui, D., Pradhan, B., Lofman, O. and Revhaug, I.: Landslide Susceptibility Assessment in Vietnam Using Support Vector Machines, Decision Tree, and Naive Bayes Models, Mathematical Problems in Engineering, 2012, e974638, 45 doi:10.1155/2012/974638, 2012

**AR:** Thank you for this comment. This citation got lost during internal revision. Now added within p.14 line 29: "(Ballabio and Sterlacchini, 2012; Catani et al., 2013; Felicísimo et al., 2013; Tien Bui et al., 2012)".
Furthermore, we changed the reference "Petschko et al., 2015" to "Petschko et al., (2016)", because this reference was recently updated (i.e. the paper got a new release year, volume number, page numbers)

**RC:** Figure 5 The color palette is not easy to distinguish landslide susceptibility in the maps. Please, use a more contrasting color palette.

**AR:** The contrast of the color palette is already high as the respective colors range from "white" (susceptibility of 0) to "black" (susceptibility of 1). Within the revised figure, we further enhanced the contrast by reducing the transparency of the maps (overlaid on a shaded relief image).
Sect. 5.2 provides an explanation on why some susceptibility maps appear uniform at slope scale (page 10, line 29) *"[…].Thus, a high positional error of the inventory (mean error: 120 m) was observed to result in susceptibility maps that were less influenced by local slope angle and slope aspect variations. Ultimately, this led to more uniformly appearing susceptibility patterns at slope scale (e.g. Fig. 5f). […]"*

**RC:** Figure 6 Indicate what is A,B,C and D in figure caption.

**AR:** Done: "*[…] for the real data sets (a, c) and the synthetic data (b, d) obtained by CV (a, b) and SCV (c, d).*"

---

## Author Response (AR2)

Dear editor,

We cordially thank you for the constructive comments. We are delighted that our paper "is now ready for publication". We revised the paper according to the suggestions ("minor/technical changes") as follows:

Editor: can you provide a picture of a landslide? Just select one from the database you considered. The readers will appreciate it.
AC: We added a photograph of a characteristic shallow landslide of the northern part of Waidhofen/Ybbs (Flysch Zone, cf. Fig. 1c) to Fig. 1 and adapted the caption accordingly.

Editor: fig.2 and 3: please shift "u5", "u10", "u20" ... "u120" in the right bottom side of the plots; now they looks very closed to the histograms.
AC: Changed as suggested.

Editor: fig. 5: please enlarge the font size of the legend.
AC: Done.

Editor: fig. 6: please enlarge the font size of the number in the matrix.
AC: Done.

Best regards,
Stefan Steger
* * *
The revised marked-up manuscript version (changes only in the figure-part) can be found below. Please note that the figures within this marked-up version are highly compressed.

[revised manuscript text omitted]

**Real data**

**Synthetic data**

Mean positional inaccuracy of the inventory (m)

**0 m: Unmodified**

(a)
AUROC
- CV: 0.845
- SCV: 0.829
- "0": 0.849

(g)
AUROC
- CV: 0.864
- SCV: 0.864
- "0": 0.865

**5 m**

(b)
AUROC
- CV: 0.841
- SCV: 0.825
- "0": 0.849

(h)
AUROC
- CV: 0.863
- SCV: 0.863
- "0": 0.865

**10 m**

(c)
AUROC
- CV: 0.826
- SCV: 0.808
- "0": 0.848

(i)
AUROC
- CV: 0.862
- SCV: 0.862
- "0": 0.865

**20 m**

(d)
AUROC
- CV: 0.793
- SCV: 0.773
- "0": 0.844

(j)
AUROC
- CV: 0.856
- SCV: 0.856
- "0": 0.865

**50 m**

(e)
AUROC
- CV: 0.764
- SCV: 0.743
- "0": 0.840

(k)
AUROC
- CV: 0.83
- SCV: 0.83
- "0": 0.864

**120 m**

(f)
AUROC
- CV: 0.686
- SCV: 0.653
- "0": 0.843

(l)
AUROC
- CV: 0.753
- SCV: 0.746
- "0": 0.859

**Landslide susceptibility**

1     0

**Inventory**

- ● Unmodified inventory ("0")
- ○ Positional inaccurate inventory

[Figure]

**Figure 5. Landslide susceptibility maps (for the location of the cutouts refer to the red boxes) of models generated with differently accurate inventories (rows) for the real data (left column) and the synthetic data (right column). Corresponding predictive performances (median AUROC for CV and SCV) and the quantitative comparison with the original landslide positions ("0") are given in the respective white boxes. The cutouts additionally show the location of the positionally erroneous inventories (white dots) used as input for the respective models and the position of the original inventory (red dot). Note that maps generated with the most inaccurate inventory (f,l) are characterized by a locally more uniform spatial pattern.**

[Figure]

[Figure]

**Figure 6. Permutation-based variable importance for the respective reference models (0 m) and models generated with positionally erroneous inventories (5 m – 120 m) for the real data sets (a,c) and the synthetic data (b,d) obtained by CV (a,b) and SCV (c,d). The relative ranking indicates that slope was the most influential predictor for each model while its estimated importance constantly decreased with an increasing inventory-based positional error (cf. Median AUROC decrease). The estimated importance of lithology slightly increases with a growing inventory-based positional inaccuracy.**